# Coffee Brews: Are They a Source of Macroelements in Human Nutrition?

**DOI:** 10.3390/foods10061328

**Published:** 2021-06-09

**Authors:** Ewa Olechno, Anna Puścion-Jakubik, Katarzyna Socha, Małgorzata Elżbieta Zujko

**Affiliations:** 1Department of Food Biotechnology, Faculty of Health Science, Medical University of Białystok, Szpitalna 37 Street, 15-295 Białystok, Poland; ewaolechno1996@gmail.com (E.O.); malgorzata.zujko@umb.edu.pl (M.E.Z.); 2Department of Bromatology, Faculty of Pharmacy with the Division of Laboratory Medicine, Medical University of Białystok, Mickiewicza 2D Street, 15-222 Białystok, Poland; katarzyna.socha@umb.edu.pl

**Keywords:** coffee, calcium, magnesium, phosphorus, potassium, sodium

## Abstract

Coffee brews, made by pouring water on coffee grounds or brewing in an espresso machine, are among the most popular beverages. The aim of this study was to summarize data on the content of macroelements (sodium, potassium, calcium, magnesium, and phosphorus) in coffee brews prepared with different methods, as well as to review the factors influencing the content of the elements. Studies from 2000 to 2020, published in the PubMed and Google Scholar databases, were reviewed. Taking into account the results presented by the authors, we calculated that one portion of coffee brew can cover 7.5% or 6.4% (for women and men) and 6.6% of the daily requirement for magnesium and potassium, respectively. Coffee provides slightly lower amounts of phosphorus (up to 2.2%), sodium (up to 2.2%), and calcium (up to 0.7% of the daily requirement for women and 0.6% for men). If coffee is drunk in the quantity of three to four cups, it can be an important source of magnesium, considering the risk of magnesium deficiency in modern societies.

## 1. Introduction

Coffee is a widely consumed drink, hence the great interest of researchers in its biochemical composition and effect on health. Coffee consumption is growing year by year. In 2017/2018, it amounted to 9,682,620 tons, while at the turn of 2020/2021, it was 9,997,680 tons [1]. There are many varieties of coffee, but most commercial coffees are called Arabica (*Coffea arabica* L.) or Robusta. Robusta is not even *Coffea canephora* var. robusta, but consists of other varieties [2,3,4]. The positive effect of coffee on the body has been discussed in several meta-analyses [5,6,7,8]. Coffee contains over one thousand bioactive ingredients [9], including: carbohydrates, lipids, proteins, chlorogenic acids, diterpenes, alkaloids, e.g., trigonelline and caffeine, free amino acids, and minerals [10]. The content of minerals varies across varieties and brews made using different methods, being lower in brews than in coffee beans [11]. Green coffee beans usually contain 3.0–5.4% of minerals on a dry matter basis [11,12]. On the other hand, roasting does not change the concentration of minerals [13]. The main component of coffee beans is potassium—it constitutes 40% of the ash [11,14].

In terms of their importance to humans, minerals can be divided into macroelements (macronutrients) and microelements (micronutrients). Elements from both of these groups are essential for the proper functioning of the human organism and have many important functions [15,16]. The demand for these substances varies significantly [17]. Both an excess and deficiency of certain elements may cause adverse health effects [15,16]. Macroelements are minerals whose content in the body exceeds 0.01% of body weight, and the demand for them is higher than 100 mg per day. Among them are: potassium, sodium, calcium, magnesium, phosphorus, sulfur, and chlorine. Micronutrients are found in human organisms in smaller amounts—less than 0.01% of body weight, and the demand for them is also lower—less than 100 mg per day. They include: iron, zinc, selenium, copper, chromium, manganese, cobalt, iodine, fluorine, molybdenum, and others. Among minerals, there are also toxic elements, such as aluminum, arsenic, mercury, lead, or cadmium [18]. These can accumulate in the body and pose a health hazard [19].

It seems that coffee can be a source of minerals in the diet, especially if consumed frequently [20]. Plants absorb nutrients from the soil, and therefore the following factors have a significant impact on the composition of coffee beans: the type of coffee, type of soil, cultivation method (including fertilization), environmental pollution (which finds its way into the soil), and production processes [21,22,23,24,25,26,27]. Moreover, it seems that the height above sea level and rainfall may also play a role [28,29,30]. On the other hand, the content of minerals in a brew is influenced by analogous factors affecting the composition of coffee beans, but also by additional variables: the method of brewing, type of water used, brewing time, the ratio of coffee to water, degree of roasting and grinding of coffee beans, or pressure in the case of methods such as coffee machine or coffee pot [13,20,31,32].

The origin of coffee seems to be the most important factor influencing the mineral composition of beans [24,33,34,35,36,37]. Countries and regions vary as to the mineral composition of the soil. It is related to the naturally occurring components and their proportions, as well as to human influence [11,24,32,34,38]. Moreover, minerals differ significantly in their extraction behavior [36,39], and it still remains unclear whether the source of the identified minerals in the coffee brew is water or the coffee itself. An additional factor is the bioavailability of individual ingredients, i.e., the amount that will actually be used by the body. This is influenced by both physiological and nutritional factors as well as food processing [40].

Coffee is a widely consumed drink, so this publication summarizes research on its mineral content to assess whether it can be considered a source of macroelements in a diet.

## 2. Materials and Methods

The study takes into account research from the years 2000 to 2020. The databases searched were Google Scholar and PubMed. The following terms were entered: ‘coffee’, ‘Arabica’, ‘Robusta’, ‘minerals in coffee brews’, ‘macroelements in coffee brews’, ‘sodium’, ‘potassium’, ‘calcium’, ‘magnesium’, ‘phosphorus’, and ‘coffee origin’. We included publications which contained most of these categories of information: the specific type of coffee, coffee origin, brewing method used by consumers, brewing time, amount of coffee and water, type of water, cup volume, temperature, pressure, type of coffee, time and degree of grinding, and the degree of roasting. The exclusion criterion was the use by the authors of a method of brewing which is not commonly used by consumers.

## 3. Results and Discussion

Table 1 and Table 2 contain the literature data obtained as part of the research review for regular and instant coffee brews, respectively. In order to compare the results by various authors, the method of presenting the content of individual minerals in brews was standardized and recalculated, and then expressed as mg/100 mL. In some cases, the results (marked as ‘*’) were presented as per 100 g to be more comparable. It would be necessary to determine the density of the brews prepared by the authors in the publications where the results were expressed as per 100 g in order to correctly assess the amounts of elements that they provide. Thus, these values were only referred to at the end of each paragraph. When the authors reported the content of macroelements per 100 g of coffee (not infusion), the content was converted into a portion of coffee adopted by the authors, and the final volume was assumed as 100 mL. If the authors did not provide a conversion for daily requirement, we made it independently.

### 3.1. Sodium Content in Coffee Brews

Sodium is a cation which dominates in the extracellular fluid of the human organism. Its role is to control the volume and distribution of the total body water, maintain the action potential of cell membranes, and participate in the conduction of nerve impulses [47]. Excess sodium in the diet, over 2 g per day, may contribute to the development of hypertension and cardiovascular diseases [48,49]. Sodium is an element that occurs commonly in food, especially processed food [50]. It has been shown that the extraction efficiency for sodium in the case of brews from roasted and ground coffee is 30.2–47.7% [36,42,46].

In the studies where the authors expressed results per 100 mL, the highest sodium content was found in Turkish brewed coffee studied by Özdestan (2013) [41]: 28.97 ± 6.35 mg/100 mL (Arabica), while the lowest was found in Turkish brewed coffee in the study by Adler et al. (2019) [46]: 0.05 ± 0.02 mg/100 mL (no species given). The brewing time was only included by Adler et al. The ratio of the amount of coffee to the amount of water used was higher in the study by Özdestan, which is consistent with the highest sodium content in this study. It can be concluded that the ratio of coffee to water could have played a role. As regards the influence of the type of water, this factor was not mentioned by Adler et al. Özdestan used ultrapure distilled water. The type of water may have influenced the extraction of the substance from the brew, but there are no studies supporting this. Water temperature was given only by Adler G.—it was 100 °C. However, since preparing Turkish coffee is based on the same principle, the temperatures were probably similar. To make traditional Turkish coffee, roasted Arabica beans are usually used, finely ground immediately before preparing the brew, and cold water, which is heated to a boil. The amount of water is measured with the cups in which the coffee will be served. Sugar is also added directly to the coffee. The amount of sugar was specified: one to two teaspoons per cup [51,52].

Ödestan used ground coffee, while Adler G. used freshly ground coffee. The time of grinding does not appear to affect the content of sodium. According to Świetlik and Trojanowska (2014) [53], ground coffee may contain more elements that are part of technological devices and find their way into it during the production process. The authors mentioned possible increases in the amount of iron and zinc. The degree of grinding of the grains was defined only in the study by Özdestan as fine ground. For Turkish-style brews, finely ground coffee is used [51,52], and such coffee was probably also used by other researchers, but due to the lack of data in other studies, this is not clear. Only the beans used by Özdestan were defined as roasted. The origin of the coffee was not given in the analyzed studies of Turkish coffee brews.

Taking into account the method of pouring water onto coffee, the sodium content in the study by Janda et al. (2020) [20] was the highest: 2.78 mg/100 mL (Arabica). Ashu and Chandravanshi (2011) [36] and Gogoaşă et al. (2016) [43] obtained lower sodium values: 0.59 mg/100 mL (no species given) and 0.12 mg/100 mL (no species given), respectively. Brewing time was the same in Janda et al. and Ashu and Chandravanshi, 5 min, while in Gogoaşă et al., it was 10 min, which did not translate into a higher content of the mineral. The amount of coffee and water used by Janda et al. was the highest: 17 g of coffee and 250 mL of water. Meanwhile, in the other studies involving ground coffee, 6 g of coffee and 150 mL of water [43] or 200 mL of water [36] were used.

In the case of ground coffee, the type of water used may have played a role. In the study by Ashu and Chandravanshi, its type was not specified. Gogoaşă et al. used distilled water, while Janda used filtered water [20,43]. There are no studies confirming an increase or decrease in sodium in coffee brews depending on the kind of water used. In addition, the type of water is important, as water is rich in various minerals and may have a different effect on the extraction of minerals in the brew, thus distorting the results. Water filtration removes certain elements [54]. However, unlike distilled water, consumers are likely to use filtered water. The solution could be to determine the individual elements in the water before making a brew.

Water temperature in the analyzed projects ranged from 92 [20] to 100 °C [36,43]. It does not seem probable that such slight variations could have affected the final sodium content in any brew. There is also a lack of research that would confirm this. Not all the researchers provided information on what specific type of coffee they used. Arabica and Robusta may differ in the content of some elements [21,22]; however, these are not significant differences. The content of elements is influenced, among other things, by soil condition and fertilization [26]. The degree of roasting of the coffee beans was not specified either, but as shown by Van Cuong et al. (2014) [13], roasting does not affect sodium content.

The degree of grinding may be important, depending on the type of coffee. It seems that the more finely ground a given substance is, the greater its extraction area [55,56]. Only Janda et al. (2020) [20] described the degree of grinding as very fine. Commercial coffees, ground at the production stage, have varying degrees of grinding—typically medium to fine. Coffee mixed with water should be ground as finely as possible [55,57]. Unfortunately, it is not known whether the other researchers followed this rule.

The origin of the beans could also play an important role. However, not all of the researchers provide such information, which makes analysis difficult. Only Ashu and Chandravanshi (2011) [36] indicated that the coffee they used came from Ethiopia. The authors tested a blend of three different brands of coffee. As shown by Habte et al. (2016), coffee from Ethiopia is not dominated by sodium, but potassium, phosphorus, and magnesium [58]. It should also be emphasized that the content of individual elements differs depending on the region [28,29,58].

In the espresso analyzed by Janda et al. (2020) [20], containing 17 g of coffee per 250 mL of brew (filtered water), sodium content was about 2.75 mg/100 mL. Espresso is characterized by a small volume—about 30 mL [59]. It follows that the coffee was probably diluted to obtain the desired volume, which could have had an impact on the final result. Different amounts of coffee and water were also used. Coffee for espresso is finely ground, and so was that used by Janda et al. The authors did not specify the origin or degree of roast of the coffees used.

The other brewing methods yielded from 2.47 (the drip method) to about 2.60 mg/100 mL (the French press) of sodium in Janda et al. (2020) [20]. These amounts are similar to those found in coffee made with the other brewing methods in the same project, which leads to the conclusion that the brewing method did not significantly affect the sodium content of the brews.

As regards the content of sodium per 100 g of brew, the highest was recorded in ground coffee mixed with water (Arabica and Robusta) in the study by Grembecka (2007) [42]: 2.52 mg/100 g. The lowest result was found in brew prepared in a coffee machine, analyzed by Oliviera et al. (2015) [32]: 0.06 mg/100 g. The high result reported by Grembecka may be due to the origin of the coffee itself.

In the case of ground coffee, the average values ranged from 0.06 to 2.52 mg/100 g of brew. These are discrepancies, possibly caused by different origins of coffee, including the use of fertilizers containing sodium. Arabica coffee in Grembecka et al. (2007) [42] contained a lower concentration of this element, respectively: 0.36 mg/100 g [42]. It was pure Arabica, albeit of different origins. On the other hand, the low result (0.08 mg/100 g) in Fercan (2016) [45] may stem from the method of brewing, the origin of the beans, and the lowest ratio of coffee to water (2 g of coffee/100 mL of water).

In the category of espresso brews, Oliveira et al. (2015) [32] noted the highest sodium content in a brew of an Arabica and Robusta blend: 0.39 mg/100 g. The lowest average sodium content was detected in Arabica brews from Central America (0.06 mg/100 g), while the lowest concentration of this element among all the samples was found in espresso from Cuba. Tagliaferro et al. (2007) [44] also tested espresso and obtained a lower result than the highest one reported by Oliveira et al.: 0.10 mg/100 g (method using washed grains). The origin of the coffee was not mentioned. In addition, the authors used different amounts of coffee and water. Tagliaferro et al. prepared a brew using 15 g of ground coffee and 400 mL of distilled water, while Oliveira et al. used 6 g of coffee, while the volume of the brew was 40 mL (distilled water). It follows that the ratio of coffee to water was significantly higher in the latter study. Tagliaferro et al. used a brewing method that is not specific to espresso coffee. Normally, this kind of brew is characterized by a small amount of water but a relatively high amount of coffee, and the brewing time is short, from 25 to 30 s [59]. The authors used a longer brewing time: 50 s. The pressure applied was also higher than that typical of a traditional espresso: 15 bar. For a standard espresso, 9 bars are usually used [60,61]. There is no literature on the effect of pressure on the extraction of minerals. It seems that it may facilitate the extraction and hence produces a much higher result. The origin of the grains, which only Oliveira et al. took into account, could also play a role. Other research methods were used, which could also have been relevant in this case. Additionally, the brew was frozen at −10 °C for 72 h. These factors could have had an impact on the concentration of the mineral in the brew.

Tagliaferro et al. (2007) [44] wanted to check whether the coffee was contaminated with soil and whether contaminants could get into the brew. The researchers divided the coffee beans into ‘washed’ and ‘unwashed’. Some of the beans were washed with cold deionized water to get rid of contaminants; the others remained unwashed. Then, the two groups of coffee were compared in terms of the presence of contaminating elements, including Fe, Th, and Sm. Sodium was not regarded as a grain contaminant. It seems that both methods of brewing and the origin of the coffee had an impact on the final sodium content of the brews. The results which both Oliveira et al. and Tagliaferro et al. obtained were lower than the highest value, reported by Grembecka et al. (2007) [42].

Instant coffee brews in Grembecka et al. (2007) [42] and Oliveira et al. (2012) [23] had 2.5 times higher sodium contents (7.14 mg/100 g) and about twice that (1.70 mg/100 g) of the highest result for ground coffee brews, respectively. In instant coffee, all the ingredients contained in the coffee powder are transferred to the brew, which may result in high mineral content [23,42,62,63,64]. The extraction efficiency for sodium in soluble coffees was 96% [42]. Instant coffee is a concentrated brew of ground coffee. The content of minerals in such a brew will depend mainly on the content in the brew of ground coffee from which the coffee powder was made [26].

The calculations were made on the basis of publications in which the results were presented per liter or milliliter of brew. The brews presented by weight of the product have been compared separately so that the comparisons are meaningful. It should be emphasized that the given coverage for the mass of the brews is very approximate, except for the results presented by Özdestan (2014). For publications in which the results were presented by weight, the calculations for discussion in terms of daily requirement were done assuming a volume of 30 mL for espresso [59]. However, for the other methods (coffee mixed with water, Aeropress, Turkish-style coffee, drip method, French press), a volume of 150 mL was adopted. The volumes given by other authors ranged from 120 to 150 mL for the methods mentioned. However, such volumes were adopted for better illustration of the obtained results [59,65,66].

According to the European Food Safety Authority (EFSA), the sodium requirement in people over 18 years of age is 2000 mg. It is lower in persons with hypertension or kidney diseases [17]. Converted to a standard cup of brew (150 mL), the highest content of sodium was found in coffee brewed following the Turkish method [41] and amounted to 43.45 mg/150 mL of brew (28.97 ± 6.35 mg/100 mL), which is 2.2% of the daily requirement. On the other hand, the least amount of sodium was detected in ground coffee mixed with water [46]: 0.07 mg/150 mL of brew, i.e., below 0.1% of the daily sodium requirement. Despite the growing popularity of alternative coffee brewing methods, it seems that the most popular method is pouring hot water on coffee and espresso. The former type of beverage provides from 0.18 [42] to 4.17 mg of sodium/150 mL of brew [20], which translates into 0.21% of the daily requirement.

Of the coffee beverages presented per 100 g, the brew with the highest sodium content in the Grembecka (2007) study would roughly cover 0.19% of the total daily requirement in the case of coffee mixed with water. The authors did not specify the type of water, which could have influenced the final result [42]. According to the calculations of the authors who assumed the requirement of 575 mg sodium/day, two cups of coffee (300 mL together, 12 g of ground coffee) will cover 0.55% of daily intake. Conversely, the lowest score in the Fercan study (2016) for Turkish coffee would provide below 0.1% of the daily intake of sodium [46]. Oliveira et al. (2012) [23], examining instant coffee, adopted a daily sodium requirement of 1500 mg, as recommended by the Commission Directive 2008/100/EC [67], in which case the consumption of one cup (2 g of coffee) covers 0.1% of the daily requirement. An espresso brew would provide about 0.83 mg/30 mL (about 2.78 mg/100 mL)—cf. Janda et al. (2020) [20]—which is below 0.1% of the daily requirement. It follows that coffee is not a significant source of sodium and its consumption is not associated with the risk of excessive intake of this element in a normal diet.

It appears that the effect of coffee on sodium excretion is related to caffeine content and its diuretic effect. Due to the fact that caffeine is an antagonist of adenosine receptors, it may affect the fluid balance in the body. It reduces sodium reabsorption in the proximal tubule and the distal part of the nephron [68,69]. Research results vary, but it can be concluded that low and moderate doses of caffeine are not diuretic, unlike high doses [70,71,72,73,74,75]. It should also be mentioned that regular consumption of coffee induces caffeine tolerance and its effect is weaker. However, tolerance may decrease as early as 12 h after discontinuing caffeine [76]. According to the European Food Safety Authority, caffeine consumption not exceeding 6 mg/kg body weight does not cause increased diuresis. However, this conclusion concerned people undertaking endurance activity [77]. Neuhauser et al. (1997) [78] showed that a daily intake of 624 mg of caffeine in the form of coffee increased urinary water excretion by 41% and sodium by 66% within 24 h. The subjects consumed no caffeine or other methylxanthines 5 days prior to the test. Moreover, the authors of the study did not specify the amount of caffeine per kilogram of body weight. Killer et al. (2014) [79], on the other hand, did not notice any effect of caffeine contained in coffee (4 mg/kg body weight) on an increase in the volume of urine excreted within 24 h of coffee consumption. Nevertheless, a slight rise in urinary sodium excretion was noted. Another study found that drinking more coffee containing 6 mg/kg body weight of caffeine resulted in a greater increase in urinary sodium excretion compared to consumption of 3 mg/kg body weight of caffeine. At 180 min, it was, respectively: 18.6 ± 13.3 and 6.8 ± 5.0 mmol of sodium. For comparison, in a control test with water, the loss of sodium was: 6.9 ± 5.4 mmol [75]. It seems that moderate coffee consumption, i.e., two to three cups a day, does not affect the fluid and sodium balance, while higher consumption of caffeine in coffee may contribute to increased diuresis and loss of electrolytes, including sodium.

### 3.2. Potassium Content in Coffee Brews

Potassium is a macronutrient that plays an important role in intra- and extracellular water distribution. It also takes part in maintaining the membrane potential and regulating the acid–base balance, supports the electrical activity of neurons and muscle cells, and plays a significant part in cell metabolism, as well as the synthesis of proteins, glycogen, and hormones [80]. This element is common and can be found, for instance, in vegetables and fruits, especially dried ones [17].

Potassium is an element that freely passes into coffee brews and is characterized by high solubility [11,39,58]. For ground and roasted coffees, the extraction efficiency of potassium is high and amounts to 72.9–88.6% [36,42,46].

Analyzing the results presented per 100 mL of coffee in the discussed papers, the highest concentration of potassium was found in Arabica coffee prepared using the Aeropress method in the study by Janda et al. (2020) [20], 154.07 mg/100 mL, while the lowest in ground coffee mixed with water in the study by Ashu and Chandravanshi (2011) [36], 37.21 mg/100 mL (average content). The latter study did not take into account the type of coffee. The remaining brews of ground coffee mixed with water contained the following amounts: about 120.00 mg/100 mL in Janda et al. (2020) [20] and 52.20 mg/100 mL in Gogoaşă et al. (2016) [43]. Among ground coffees, Janda et al. recorded the highest potassium content using, at the same time, the highest coffee/water ratio. It follows that this factor could have played an important role here. However, the type of water could have distorted the results to some extent, as mentioned earlier. It was identified as filtered in Janda et al., distilled in Gogoaşă et al., and remained unspecified in Adler et al.

The temperature of water ranged from 92 °C in Janda to 100 °C in the other two studies; however, there are no literature data to assess the effect of this difference on potassium content. The type of roasting in studies under consideration is not always specified. According to van Cuong et al. (2014) [13], an increase in the degree and temperature of roasting may cause an increase in the content of potassium, but this seems to be related to a lower water content in roasted beans. Information on the degree of grinding was only provided by Janda et al. as very finely ground. Only Ashu and Chandravanshi [36] mentioned the origin of the coffee: a blend of coffees from Ethiopia was used. Origin is an important factor that could have influenced the differences. According to Nieder et al. (2018) [15], nowadays, soils are often deficient in potassium, so the element has to be regularly supplied, for example in the form of fertilizers.

Janda et al. (2020) [20] found about 100 mg/100 mL of potassium in espresso coffee (Arabica). In their study, a prolonged espresso could have been used, or the coffee might have been diluted to the desired volume, as mentioned earlier. That could have lowered the concentration of the mineral in the brew.

Turkish brewed coffee contained the following amounts of potassium: 99.20 ± 17.94 mg/100 mL in Adler et al. (2019) [46] and 57.03 ± 9.73 mg/100 mL in Özdestan (2013) [41]. The differences may result from the different ratios of the used coffee to the amount of water, different types of water, degrees of roasting and grinding, origin and, to a lesser extent, types of coffee. Higher coffee to water ratio was noted by Özdestan, but it did not increase the potassium content. The type of water in Adler et al. was not specified, but it could have had an impact on the final result. Information on the degree of grinding of the beans (finely ground) was only provided by Özdestan as regards coffee used for Turkish-style brew. Adler et al. did not record the type of grounds they used. Furthermore, the roast and origin of the coffee, which may also have played a role, were not specified in either study.

In Janda et al. (2020) [20], coffee brewed with other methods contained the following amounts of potassium: the drip method—about 140.00 mg/100 mL; the French press method—about 88.74 mg/100 mL. Comparing all the methods used in the above study, the Aeropress method was the most efficient as regards the extraction of potassium; the lowest extraction was noticed in the case of the French press method.

As for potassium content per 100 g of brew, espresso had the highest concentration of the element: 300.85 mg/100 g of brew (unwashed Arabica) in Tagliaferro et al. (2007) [44], while the lowest average content was found in ground coffee (no species given) mixed with water in Grembecka et al. (2007) [42]: 82.14 mg/100 g. Comparing the espresso brew described by Tagliaferro et al. [44] to that analyzed by Oliveira et al. [32], it can be concluded that the differences are significant: 300.85 vs. 167.44 mg/100 g. This may be due to both the origin and the differences in the methods of brewing. In Oliveira et al., the highest potassium content was detected in coffee from Kenya and Brazil, while the lowest concentration of this element was detected in coffee from Mexico [32]. Debastiani (2019) [39] also noted that potassium content might vary depending on the brand of coffee. Moreover, the authors emphasize that a high content of potassium in brews may also result from the high solubility of this element. The high score of coffee from Brazil, apart from the origin of the coffee itself, could have been influenced by the use of another type of coffee, in the form of capsules, which could have differed as to the degree of grinding. The brew prepared from unwashed beans by Tagliaferro et al. had a slightly higher potassium content (300.85 mg/100 g) than the one made from washed beans (294.91 mg/100 g), although the authors did not mention this difference and focused on other elements that are a sign of soil contamination.

Comparing the obtained contents in the brews of ground coffee to those found in soluble coffee, it can be noticed that the average potassium contents were lower than the highest result reported by Tagliaferro et al. (2007) [44], respectively: 283.99 [23] and 50.60 mg/100 g [44]. On the other hand, the maximum result in Oliveira et al. (2012) [23] was higher. It follows that instant coffee may contain more potassium than brews of individual ground coffees. This may be due to the complete dissolution of the coffee powder and thus the release of the ingredients into the brew, as mentioned above. The release of potassium in the case of soluble coffee can be as high as 98.8% [42]. Moreover, the higher potassium content in soluble coffee noted by Oliveira et al. compared to Grembecka et al. (2007) [42] most likely resulted from the ratio of coffee to water, higher in the former paper, as well as the origin of the coffee from which the coffee powder was produced.

Persons above the age of 15 need 3500.00 mg/d of potassium. The requirement is higher for breastfeeding women and people with hypertension, and lower for those suffering from kidney diseases [17]. The highest content of potassium, expressed per serving, was detected by Janda et al. (2020) [20] in Aeropress coffee: 231.11 mg/150 mL of brew, i.e., 6.6% of the daily potassium requirement. The lowest concentration was found by Ashu and Chandravanshi (2011) [36] in coffee mixed with water: 55.82 mg/150 mL of brew, 1.59% of the daily requirement. In the case of roasted coffee mixed with water, a cup of coffee may cover from 1.59% [36] to 5.14% [20] of the daily norm for this element, while a cup of espresso provides about 4.29% of the daily intake [20].

The highest result obtained for espresso among brews presented per mass of brew would provide 12.9% of the daily requirement for this element [44]. On the other hand, the lowest result obtained by coffee poured with water in Grembecka et al.’s study (2007) [42] would cover about 3.2% of the daily requirement for this element. This is the value obtained by the authors’ calculations (two cups of coffee, 12 g of coffee product, daily intake: 3500 mg of potassium). Oliveira et al. (2012) [32] adopted a different norm for potassium: 2000 mg/day, so one cup of instant coffee (2 g) would cover 4.75% of the daily requirement. This means that coffee can provide potassium and, given an average consumption of three to four coffees a day, it can be a source of this element.

The influence of coffee consumption on potassium balance, as in the case of sodium, depends on the dose as well as tolerance to caffeine [70,71,72,73,74,75,76]. In Seal et al. (2017) [75], it was found that consumption of coffee providing 6 mg/kg of body weight of caffeine causes increased loss of potassium in the urine, in contrast to coffee providing 3 mg/kg of body weight of caffeine. Correspondingly, at 180 min, potassium excretion in the case of low caffeine consumption was 4.5 ± 2.8 mmol, and in the case of high caffeine consumption, 8.5 ± 5.7 mmol. In the control sample with water, potassium excretion was 4.8 ± 4.0 mmol [75]. Neuhäuser et al. (1997) [78] also noted increased potassium excretion (28% within 24 h of consumption) with a high caffeine intake of 642 mg/day. The authors of this study, as mentioned before, did not specify the dose per kilogram of body weight. Thus, it seems that moderate coffee consumption will not significantly increase the loss of potassium in urine. Tolerance to caffeine, and thus its reduced effect, should also be taken into account in people who regularly consume this substance. Drunk in moderate amounts, coffee can be a source of potassium, especially among regular consumers.

### 3.3. Calcium Content in Coffee Brews

The main role of calcium is to maintain bone mineralization. As much as 99% of the element is bound in the bones in the form of calcium–phosphate complexes (hydroxyapatite). The remaining part is unbound calcium, which acts, inter alia, as an enzyme activator, is involved in the transmission of nerve signals, muscle contraction, and blood clotting [17,81]. The primary sources of calcium are milk and dairy products, as well as fish, nuts and seeds, dark green vegetables, calcium-rich water, and calcium-fortified food, for example, soya drinks [82].

A high content of calcium in water makes it difficult to release this element from ground coffee. The content of calcium in a brew may be lower than in the tap water with which coffee was made, while in the case of mineral water, these concentrations are comparable. This is due to the formation of complexes of calcium from water with polyphenols or other organic ingredients [83,84]. Precipitation of CaCO_3_ is also possible when water is highly mineralized [84]. In the study by Stelmach et al. (2013) [31], tap water did not cause leaching of calcium, but a higher calcium content was found in distilled water than in mineral water. According to Debastiani et al. (2019) [39], calcium is one of the minerals in contact with which coffee acts like a ‘sponge’. This means that coffee components, including polyphenols, form complexes with calcium.

The highest calcium content (expressed as mg/100 mL) was obtained in the brew of coffee mixed with water in the study by Gogoaşă et al. (2016) [43]: 3.49 mg/100 mL, while the lowest was found in the Turkish brew analyzed by Adler et al. (2019) [46]: 1.38 ± 0.29 mg/100 mL. The other brews made by pouring hot water on ground coffee contained slightly lower values than the sample studied by Gogoaşă et al.—about 2.30 mg/100 mL (Arabica, no origin) in Janda et al. (2020) [20], and 1.62 mg/100 mL in Ashu and Chandravanshi (2011) [36]. The coffee to water ratios varied from 6 g of coffee/200 mL of water in Ashu and Chandravanshi, through 6 g/150 mL in Gogoaşă et al., to 17 g/250 mL in Janda et al. Despite the highest coffee/water ratio, Janda et al. did not obtain the highest calcium content, which indicates that this is not a significant factor. Brewing time in Gogoaşă et al. (10 min) was twice as long as that in Janda et al. and Ashu and Chandravanshi, which could have influenced calcium concentration. In the brew made by pouring water on coffee which contained the lowest amount of calcium, the type of water was not specified, which could have influenced the calcium content. Distilled and filtered water was used in the remaining studies. It was also shown that filtering reduces the amount of calcium in tap water [54,85]. However, Gogoaşă et al. obtained a higher concentration of this element despite the use of distilled water. This might mean that a different mechanism worked there.

The temperature of water in the three above-mentioned studies was 92 [20] and 100 °C [36,43]. As shown by Stelmach et al. (2013) [31], a water temperature of 60–80 °C does not change the extractability of calcium for brew, while the use of water at a temperature of 90 °C increases calcium content in a brew by 20% and reaches its maximum value at 100 °C. It seems that the use of boiling water could have resulted in greater leachability of calcium in the brew. All compared brews were prepared from coffee ground at the production stage, and only Janda et al. [20] defined the degree of grinding as finely ground. The degree of roasting is not provided, but its increase may have had a positive effect on the extraction of calcium into the brew, as demonstrated by van Cuong et al. (2014) [13]. The highest calcium extraction was noticed at the roasting temperature of 250 °C [13]. Of the poured-over coffees, origin was only given by Ashu and Chandravanshi, who obtained the lowest result using the same brewing time. As has been shown, Ethiopian coffees, depending on the region of cultivation, may be characterized by a high content of calcium [58]. Endaye et al. (2020) [37] claimed that calcium can be considered an ingredient that differentiates the geographical origin of coffee.

In the case of coffee brewed in an espresso machine, Janda et al. obtained 2.57 mg/100 mL of calcium. The remaining brews in Janda et al. had lower contents of this element. It follows that this method enhances the extraction of calcium.

Among the ground coffee brews (results expressed per 100 g of brew), the highest calcium content was found in the brew (Arabica) by Stelmach et al. (2013) [31]: 43.23 mg/100 g of brew. The lowest result was obtained by Oliveira et al. (2015) [32], also for coffee made in an espresso machine. The average calcium content was 1.11 mg/100 g of brew, with the lowest concentration of calcium in espresso from Honduras. The aim of the study by Tagliaferro et al. [44] was to identify elements that can be markers of soil contamination (8.31 mg/100 g in unwashed beans and 7.81 mg/100 g in washed beans). However, calcium was not recognized by the authors as a contaminant. It seems that the preparation method could have had an impact, including the significantly higher pressure and the origin of the coffee. In Oliveira et al., brews made from African beans had the highest mean concentration of calcium. The highest result among them was detected in espresso (Arabica) from Ethiopia. The brews differed significantly depending on the origin. According to Feleke et al. (2018) [28], coffees from different regions of Ethiopia are characterized by a high concentration of calcium, right after potassium. Moreover, the authors emphasize that coffee can supplement the diet with this element.

Taking into account brews of ground coffee mixed with water, Grembecka et al. (2007) [42] obtained calcium values lower (by about 6.4 times and 8.56) than Stelmach et al. (2013) [31]: 6.78 mg/100 g for Arabica and 5.05 mg/100 g for Arabica and Robusta ground coffee [42]. Stelmach et al. noted the following concentrations of this element: 43.23 for an Arabica brew and 37.97 for a blend of Arabica and Robusta. These differences may be due to the origin of the coffee beans, brewing time or type of water, which was not specified in Grembecka et al.’s study [42].

Instant coffee brews contained less calcium than the espresso brew in Tagliaferro et al. (2007) [44]—1.61 mg/100 g in Oliveira et al. (2012) [23] and 6.96 mg/100 g in Grembecka et al. (2007) [42]. On the other hand, ground coffee brews in Grembecka et al. [42] obtained lower values (6.78 and 5.05 mg/100 g) compared to soluble coffees (6.96 mg/100 g). If we compare ground coffee to soluble coffee brews in the same study using the same variables, soluble coffee also proved to have a higher concentration of calcium. The release of calcium into the brew for this type of coffee was 96.5% [42].

The daily calcium requirement of people over 25 years of age is 750 mg/per day for women and 950 mg/per day for men [17]. Given the lowest and the highest result, one cup of coffee can provide from 2.07 (Turkish brewed coffee [46]) to 5.24 mg/150 mL (ground coffee mixed in water [43]). Converted to the daily requirement for calcium, the brews would cover from 0.3 to 0.7% of the daily requirement of women and from 0.2 to 0.6% of the daily requirement of men. As regards the calcium content obtained using the simplest brewing method in all the studies discussed (in milliliters of brew), coffee mixed with water would cover 0.3–0.7% of the daily calcium requirement of women and 0.3–0.6% of the demand of men [36,43]. For comparison, the highest result for the brew presented per mass (espresso, about 30 mL) would cover 0.3% of the daily requirement of calcium for women and for men [44]. On the other hand, the lowest result on the mass of brew was obtained by espresso in the Oliveira study (2012). According to the authors, a brew (5–6 g of coffee) would provide about 0.1% of the calcium requirement. They took the standard of 800 mg of calcium per day [23]. According to Oliveira et al. (2012) [23], consumption of one cup of instant coffee (2 g of coffee) would provide 0.2% of the calcium requirement (norm of daily intake: 800 mg). On the other hand, an espresso brew could provide 0.77 mg/30 mL of calcium, i.e., 0.1% of the demand of women and below 0.1% of the demand of men (Janda et al. (2020) [20]). However, according to Oliveira et al., an espresso brew would cover 0.1–0.3% of the daily norm (800 mg) of this element.

The amount of calcium in coffee is negligible. Moreover, it has been shown that the caffeine contained in coffee can negatively influence the intestinal absorption of calcium and increase its excretion in urine, which in turn has a negative effect on the skeletal system. A meta-analysis from 2014 showed a 14% increase in the risk of bone fractures in women who consumed eight cups of coffee a day [86], while moderate caffeine consumption (up to 400 mg/day) did not adversely affect calcium metabolism or the incidence of fractures [87]. Another study from 2016 even found a positive effect of coffee consumption and a negative correlation with the occurrence of osteoporosis in 4,066 Korean postmenopausal women (mean age 62.6 years) [88]. Yet another study showed a protective effect of coffee drinking on bone health in 2,682 Taiwanese persons based on the T-Score [89]. The genetically determined metabolism of caffeine may also play a role here [90]. However, this requires further research. It can be concluded that coffee is not a source of calcium, but does not pose a threat to bone health if consumed in moderation. In addition, the possible negative impact can be mitigated by the provision of an adequate amount of calcium in the diet.

### 3.4. Magnesium Content in Coffee Brews

Magnesium plays an important role in the body as a cofactor of more than 300 enzyme systems, and thus participates in the regulation of blood pressure, muscle contraction, myocardial excitability, DNA and RNA protein synthesis, as well as in neuromuscular conduction and insulin metabolism [91,92]. Its deficiency is common in modern societies [93]. The causes include a low-magnesium diet, stress, high alcohol consumption, endocrine diseases (diabetes, metabolic syndrome), the use of certain medications (including proton pump inhibitors, cardiac glycosides and diuretics), and congenital magnesium transport disorders [93,94]. It has been shown that magnesium deficiency increases inflammation in the body, and thus increases the risk of cardiovascular diseases, metabolic syndrome and diabetes [95]. A meta-analysis from 2013 showed that dietary magnesium intake and its concentration in blood serum are inversely related to the risk of developing cardiovascular incidents. The greatest decrease in risk was noticed when increasing magnesium intake from 150 to 400 mg/day [96]. Therefore, commonly consumed foods that can provide this nutrient are essential. The sources of magnesium include: green vegetables (thanks to chlorophyll), nuts, seeds, unprocessed grain products, such as groats, medium-level legumes, fruits, meat, and fish [97]. Coffee could also be a source of this element. Magnesium is an ingredient that is freely released into brews. Its extraction efficiency for roasted and ground coffees is 45.8–89.7% [36,42,46].

The highest content of magnesium was found in the brew of Turkish brewed coffee in Özdestan (2013) [41]: 14.92 mg/100 mL, while the lowest was found in poured-over coffee in Gogoaşă et al. (2016) [43]: 2.15 mg/100 mL. In the category of Turkish brewed coffee, Özdestan obtained 100% more magnesium than Adler et al. (2019) [43], who received 7.00 ± 1.37 mg/100 mL of the brew. These differences may be due to the coffee/water ratios—Özdestan used a lower amount of water. The coffee/water ratio is thus consistent with the obtained magnesium values. The type of water was not taken into account by Adler et al., while in Özdestan’s study it was distilled water. This could have had an impact on the concentration of magnesium. Moreover, no information on the origin or degree of roasting of the coffee is included, although they seem to be very important factors. Turkish-style brews are typically made from finely ground coffee, but only Özdestan specified the grinding degree of coffee as finely ground, which is important for the extraction of brewing ingredients.

As regards ground coffee mixed with water, the highest content of magnesium was found in an Arabica brew made by Janda et al. (2020) [20]: about 10.70 mg/100 mL. The brew prepared by Ashu and Chandravanshi (2011) [36], containing 2.83 mg/100 mL, had slightly more magnesium than that analyzed by Gogoaşă et al. (2016) [43], but still almost four times less than reported by Janda et al. The ratio of coffee to water in Janda et al., however, was far higher, as mentioned earlier (17 g of coffee/250 mL of water) than in Ashu and Chandravanshi (6 g/200 mL) and Gogoaşă et al. (6 g/150 mL). The brewing times in the analyzed studies were 5 and 10 min. Despite the shorter brewing time, Janda et al. achieved a higher content, which means that a different factor played an important role there. Stelmach et al. (2013) [31] noted that an increase in the brewing time from 5 to 10 min increased magnesium concentration in the brew by 28%. The temperature of water ranged from 92 °C in Janda et al. to 100 °C in Ashu and Chandravanshi and Gogoaşă et al. Stelmach et al. found that the concentration of magnesium was 20% higher in the brew made with the use of water at a temperature of 100 °C compared to lower temperatures. It can be concluded that the coffee used by Janda et al. contained a higher amount of magnesium, and hence the slightly lower temperature did not matter. The same researchers also reported that tap water contributed to an increase in magnesium content in the brew by about 30%, while brews made with mineral and distilled water had lower and comparable contents of this element, respectively [31]. Janda et al. used filtered water and it seems that filtering may reduce the concentration of magnesium in water, which may have influenced the final result. Ashu and Chandravanshi did not specify the type of water, which implies the use of tap water. However, they obtained a lower concentration of magnesium than Janda et al. This would not confirm the conclusions drawn by Stelmach et al., but other factors certainly played a role. The origin of coffee could also have been relevant, but it was not identified in two studies: Janda et al. and Gogoaşă et al. Moreover, according to Habte et al. (2016) [58], Ethiopian coffee is characterized by a relatively high content of magnesium. The concentration of magnesium in coffee prepared in an espresso machine was about 8.50 mg/100 mL in Janda et al. Among the other methods in the same study, a coffee brew made using the Aeropress method proved to have the highest concentration of magnesium: 11.63 mg/100 mL, and the lowest was detected in coffee made in a French press: 7.72 mg/100 mL. It seems, therefore, that this method enhances the extraction of magnesium in the brew. This may be due to the fact that the pressure in the Aeropress method was 2–4 bar. In the case of a coffee machine, however, the pressure was higher and amounted to about 9 bar. So, another factor probably played a role.

Among the results presented per 100 g of brew, the highest magnesium content was obtained in ground coffee (Arabica and Robusta) mixed with water in Stelmach et al. (2013) [31]: 198.47 mg/100 g, while the lowest was found in Turkish coffee in Fercan et al. (2016) [45]: 2.15 mg/100 g. Such differences may result not only from the method of brewing, but also from the origin and the ratio of coffee to water, which was higher in the former study. Comparing the study by Stelmach et al. to the research by Grembecka et al., where water was also poured on coffee, Grembecka et al. obtained lower results (12.60 vs. 198.47 mg/100 g). The values in both studies significantly fluctuated, depending on the type of coffee. The ratio of coffee to water was higher in Grembecka et al., but they did not specify the type of water, which may play an important role in the extraction of the mineral. As mentioned earlier, Stelmach et al. did not specify the origin of the coffee they used, while Grembecka et al. did not mention the brewing time. It seems that these are important factors that may affect the leachability of elements.

The magnesium content in espresso analyzed by Oliveira et al. (2015) [32] varied depending on the origin of the coffee. The highest average concentration of magnesium was found in brews of coffee from South America and the lowest was found in Central American coffee brews. Taking into account individual countries, the highest content of this element was found in espresso from Brazil, while the lowest was found in espresso from Cuba. According to Debastiani et al. (2019) [39], coffees from Brazil tend to contain a lot of magnesium, at a level just below that of potassium. However, coffees from Ethiopia may also have similar characteristics [98].

Instant coffee was proved to have a lower mean content of the mineral than the highest result for ground coffees: 29.09 mg/100 g in Oliveira et al. (2012) [23] and 6.80 mg/100 g in Grembecka et al. (2007) [42]. The highest concentration of magnesium was noted in the former of these studies. Magnesium extraction in the case of soluble coffee, similarly to other elements, is almost complete, i.e., 93.6% [42].

According to the EFSA, the magnesium requirement of healthy people over the age of 25 is 300 mg/day for women and 350 mg/day for men [17]. Taking into account the lowest and highest amounts of the element, expressed per 100 mL, reported by the authors of the analyzed studies, the content per cup (150 mL) varies from 3.22 mg for a ground coffee brew mixed with water [43] up to 22.38 mg for Turkish coffee [41], which covers 1.1–7.5% of the daily magnesium requirement for women and 0.9–6.4% for men. Among the brews prepared by pouring water on ground coffee, a cup (150 mL) can provide 3.22–16.05 mg of magnesium, covering 1.1–5.4% of a woman’s requirement and 0.9–4.6% of a man’s requirement [20,36,43]. In the case of instant coffee, according to Oliveira et al. (2012) [23], one cup of instant coffee (2 g of coffee) would cover 2.6% of the daily magnesium requirement (the norm accepted by authors is 375 mg/day). Espresso coffee in Janda et al. would provide 2.55 mg/30 mL of magnesium, which covers 0.9% of the requirement of women and 0.7% of the demand of men. Comparing the results presented per milliliter/liter of brew to those expressed by mass, the highest result in Stelmach et al.’s study (2013) [31] (198.47 mg/100 g, coffee poured with water) would cover about 100% of the magnesium daily intake for women—which seems to be a very overstated value. On the other hand, the lowest result for Turkish coffee would cover about 1.1% and 0.9% of the daily requirement for women and men, respectively [46].

In the study by Oliveira et al., the authors adopted the standard of 375 mg of magnesium/day, which means that 1.4–2.2% of the demand for this element would be met by consuming one espresso (5–6 g of coffee). What stems from this is that coffee can be one of the sources of magnesium, particularly Turkish coffee, coffee mixed with water, or instant coffee, especially when it is consumed in moderate to larger quantities.

There are no studies to assess the effect of coffee consumption on urinary magnesium excretion. This is probably related to the dose and caffeine tolerance. A study by Massey et al. (1984) [99] assessed the effect of coffee and tea (caffeine content 0–300 mg) on the excretion of minerals, including magnesium, in 12 female students. An increase in magnesium excretion was noted during the three-hour measurement. In another research project by the same authors, an identical effect of caffeine was noticed in 15 healthy men [100]. A study from 1994 (Kynast-Gales et al. [101]) assessed the effect of caffeine alone on urinary magnesium excretion. The consumption of two doses of caffeine at 3 mg/kg of lean body mass increased the excretion of magnesium. However, this effect was noticeable only after the second dose of caffeine, i.e., after the delivery of 6 mg/kg of lean body mass. The content of magnesium detected in urine was 0.16 mmol in total 24 h after coffee consumption (with the content of this element in the diet amounting to 12.7 mmol).

In Saito et al. (2011), coffee was recognized as one of the sources of this element [102]. The study group consisted of nineteen healthy and non-smoking people aged 21–27 (11 women and 8 men). For 2 weeks, they consumed 450 mL of coffee (18 g a day), at the same time following a proper diet, abstaining from alcohol, and refraining from strenuous exercise. After that period, their magnesium levels surprisingly increased by 4.3%. In another study, where the bioavailability of minerals was assessed in an in vitro model with a gastrointestinal juice solution, it was shown that magnesium had the highest bioavailability: 62% [103]. The above results indicate that coffee can be an important source of magnesium.

### 3.5. Phosphorus Content in Coffee Brews

Phosphorus is involved in various physiological processes. About 85% of this element is found in the bones and teeth, 14% in soft tissues, and only 1% in the extracellular fluid. Its role is to participate in the mineralization of bones and teeth, regulate the acid–base balance, as well as regulating cell signaling and the cell’s energy cycle. Phosphorus is closely related to calcium homeostasis [17]. It is widely distributed in foods, especially highly processed ones. Natural sources of phosphorus include milk and dairy products, meat, poultry, fish, grain products, and legumes [104]. Phosphorus is an element that dissolves relatively well in water and becomes a brew [39]. The extraction efficiency for this element for roasted and ground coffee is on average 45.85% [42].

Only Janda et al. (2020) [20] took into account the content of phosphorus in an Arabica brew (results expressed as mg/100 mL). The highest concentrations of this element were noted for the Aeropress method: 8.16 mg/100 mL, while the lowest concentration was found for coffee brewed in a French press: 4.96 mg/100 mL. The amounts of coffee and water used differed slightly: 18 g/250 mL for Aeropress and 17 g/300 mL for the French press method. If we assume that the volume of the brew was similar to the amount of water, the coffee/water ratio was higher in the Aeropress method, hence probably the higher concentration of the element. An additional factor in the case of the Aeropress could also have been the pressure of 2–4 bar. Phosphorus content in the espresso was about 5.6 mg/100 mL, which may have been due to the ratio of coffee to water, which was 17 g/250 mL of the brew. The authors did not use the traditional volume for espresso, which could have biased the results. The higher pressure (9 bar) did not play a significant role. The concentration of phosphorus detected in coffee made with the drip method was similar to that in the simple brew—about 7.2 mg/100 mL. The method of brewing can therefore be of considerable importance.

The highest content of phosphorus per 100 g of brew—40.39 mg/100 g—was found by Oliveira et al. (2015) [32] in an Arabica espresso made from coffee from Kenya (Africa), while the lowest average phosphorus concentration was found in a brew of ground coffee mixed with water in Grembecka et al. (2007) [42]: 13.68 mg/100 g. These differences may result from the method of brewing, as pressure in the machine may have been an important factor. The coffee/water ratio—much greater in the study by Oliveira et al.—could also have had an impact. In addition, the type of water (not specified by Grembecka et al.), the degree of roast and grinding (not defined in either study), and the origin (not specified by Grembecka et al.) may also have played a role.

Comparing individual espresso brews in the study by Oliveira et al., it can be noted that the highest average phosphorus content was found in South American coffees, while the lowest in Central American ones, with the lowest of all in a brew made from Cuban coffee.

Instant coffee brews contained, on average, lower phosphorus concentrations than ground coffee brews: 24.78 mg/100 g in Oliveira et al. (2012) [23] and 7.76 mg/100 g in Grembecka et al. (2007) [42]. However, with regard to the minimum and maximum values, the highest concentration was found in a brew of instant coffee in Grembecka et al. [42].

According to the EFSA, the demand for phosphorus is 550 mg/day [17]. Coffee can provide from 7.45 mg/150 mL of phosphorus when brewed with the French press method to 12.24 mg/150 mL in Aeropress coffee, which is 1.4–2.2% of the daily requirement for this element [20]. A brew of ground coffee with water would provide 11.25 mg of phosphorus per cup, or 2.1% of daily intake [20], while one cup of instant coffee (2 g of coffee), according to Oliveira et al., covers 1.1% of daily intake (accepted standard: 700 mg). The espresso brew (30 mL) from the study by Janda et al. would cover 0.3% of the demand. In contrast, a cup of espresso (5–6 g of coffee) from Oliveira et al. would meet as much as 2.8–7.2% (reference to other standards) of the demand for phosphorus. The authors assumed a daily phosphorus requirement of 700 mg and did not give information on the volume of the brew. For comparison, after conversion, the highest result obtained by espresso in the Oliveira et al. study (2015) would cover about 0.2% of the daily requirement for this element. However, the lowest result expressed per mass in the Grembecka et al. study (2007) for coffee with water poured over it would cover about 2.0% of the daily intake (assuming the phosphorus requirement of 650 mg/day, two cups of coffee). Coffee can therefore provide a certain amount of phosphorus, but in a significantly lower amount than calcium, which can affect the calcium–phosphate balance. Since phosphorus is ubiquitous in food, people should consume coffee with milk and include good sources of calcium in their diet.

There are no human studies on the effects of coffee on phosphorus excretion in the urine. A 1985 study showed that coffee with 0–300 mg of caffeine did not increase urinary phosphorus excretion in 15 healthy men. In the same study, however, an enhanced effect on the excretion of magnesium, calcium, sodium, and chloride was noted [100]. Studies on rats have also shown that coffee increases the excretion of phosphate in the urine [105,106]. It seems that this effect, similarly to the other elements, is dependent on the dose and tolerance level. However, this requires further research.

To sum up, coffee brews differ in the content of individual macroelements. Factors that can influence this are the brewing method used, including brewing time, amount of coffee and water, cup volume, type and temperature of water, pressure (with some methods), but also the type of coffee, degree and time of roasting of coffee beans, degree of grinding, as well as the origin of coffee, including, but not limited to, soil type, cultivation method, environmental conditions, and production processes. It has been noted that ground coffee brews (results given in mg/100 mL) can provide a certain amount of magnesium (1.1–7.5% daily intake for women and 0.9–6.4% daily intake for men) and potassium (1.6–6.6% daily intake, maximum: 12.9%), and slightly lower amounts of phosphorus (1.4–2.2% daily intake, maximum: 7.2%), sodium (below 0.1–2.2% daily intake), and calcium (0.3–0.7% daily intake for women and 0.2–0.6% for men). Soluble coffees analyzed in the studies under discussion tended to have a higher average content of individual elements (except for potassium and calcium) than brews of ground coffee mixed with water in the case of brews given in milligrams per 100 g. On the other hand, when comparing these two types of brews in the same study, where the same variables were used, instant coffee had higher concentrations of all examined macronutrients. It seems that the origin of coffee, the soil from which the coffee shrub draws certain substances, and the method of cultivation, including the use of fertilizers, have the greatest influence on the content of minerals in coffee brews. Not all authors provided information on the origin the coffees they used. There was no particular trend in the effect of the different brewing methods on the mineral content of coffee brews. This is because some minerals behave differently depending on the method. Since the mineral content of coffee brews is influenced by a large number of factors, reliable comparison is difficult. However, it should be emphasized that coffee brews can be a source of macroelements in the diet of people who drink larger amounts of coffee, provided that they do not consume more than 400 mg caffeine per day (the norm for adults) due to the possible diuretic effect [77].

Moreover, macroelements such as potassium, sodium and phosphorus are commonly found in food, therefore no additional sources are sought. On the other hand, deficiencies of calcium and magnesium are observed more often. It therefore seems that coffee can be considered one of the sources of magnesium and does not cause its loss. In addition, it is worth emphasizing that the content of elements in the coffee brew will largely depend on the content and type of water. Most of the studies used water which is devoid of minerals, such as distilled water. By contrast, consumers tend to use tap, filtered or bottled water. In studies where a different type of water was used, e.g., filtered or tap water, it is unclear what the main source of minerals was. For this purpose, research should be deepened and the minerals in the water should be determined.

The limitations of this literature review include: different methods of presenting the research results by the authors (e.g., per 100 mL or per 100 g), failure to indicate all significant factors in the publications (coffee type, origin, amount of coffee, amount of water, final volume, fragmentation, method of roasting, content of minerals in water). Further research should take into account all the important factors influencing the macronutrient content of coffee brews.

## 4. Conclusions

Coffee brews, consumed in the amount of three to four cups of coffee a day, can be considered a source of magnesium and potassium in human nutrition. Magnesium in particular is an element whose deficiency seems to be frequent in modern society, due to factors including stress. Due to the fact that the type of water used to prepare the coffee brew is important, it is still unclear what has the greatest impact on the mineral content of the coffee brew. This issue requires detailed studies that also assess the amount of minerals in the water.

## Figures and Tables

**Table 1 foods-10-01328-t001:** The content of macroelements per 100 mL of coffee brew.

Content Av. ± SD (mg/100 mL or 100 g)	Method of Brewing	Time (Min.)	Coffee (g)	Water (mL)	Cup Volume (mL)	Type of Water	Pressure (Ba)	Temperature of Water (°C)	Species	Degree of Roasting	Type of Coffee	Origin	Methods of Analysis	References
Na														
28.97 ± 6.35	Turkish coffee	nd	5	65	nd	up dw	nd	nd	A	R	fine ground	nd	HR-CS-FAAS	[41]
~2.78	Pouring water	5	17	250	nd	fw	nd	92	A	R	very fine ground	nd	ICP-OES	[20]
~2.75	Coffee machine	nd	17	nd	250	fw	9	92	A	R	very fine ground	nd	ICP-OES	[20]
2.60	French press	5	17	300	nd	fw	1–2	92	A	R	medium ground	nd	ICP-OES	[20]
~2.54	Aeropress	2	18	nd	250	fw	2–4	93	A	R	coarse ground	nd	ICP-OES	[20]
2.52 *	Pouring water	nd	6	150	nd	nd	nd	nd	A&R	R	ground	nd	FAAS	[42]
2.47	Drip method	2.5	18	300	nd	fw	nd	92	A	R	medium coarse ground	nd	ICP-OES	[20]
0.59	Pouring water	5	6	200	nd	nd	nd	100	nd	nd	ground #	Ethiopia	FAAS	[36]
0.39 *	Coffee machine	nd	5	40	nd	dw	nd	nd	A&R	R	ground (capsules)	Asia (India)	HR-CS-AAS	[32]
0.36 *	Pouring water	nd	6	150	nd	nd	nd	hot	A	R	ground	India, Australia, Tanzania, Peru, Cuba, Timor, Zambia, Honduras, Indonesia	FAAS	[42]
0.27	Coffee machine	nd	5	40	nd	dw	nd	nd	A	R	capsules—ground	South America (Colombia, Brazil)	HR-CS-AAS	[32]
0.21 *	Coffee machine	nd	6	40	nd	dw	nd	nd	A	R	ground	Asia (China, Timor)	HR-CS-AAS	[32]
0.12 *	Coffee machine	nd	6	40	nd	up dw	nd	nd	A&R	R	ground	Africa (Kenya)	HR-CS-AAS	[32]
0.12	Pouring water	10	6	150	nd	dw	nd	100	nd	nd	ground coffee	nd	FAAS	[43]
0.12 *	Coffee machine (freeze-dried beverage)	0.83	15	400	300	diw	15	94	A	nd	fine fresh ground (washed)	nd	INAA	[44]
0.11 *	Coffee machine	nd	6	40	nd	up dw	nd	nd	A	R	ground	Africa (Mussulo and Ethiopia)	HR-CS-AAS	[32]
0.10 *	Coffee machine (freeze-dried beverage)	0.83	15	400	300	diw	15	94	A	R	fine fresh ground (washed)	ns	INAA	[44]
0.08 *	Turkish coffee	nd	2	100	nd	dw	nd	nd	A	R	nd	nd	ICP-OES	[45]
0.07 *	Coffee machine	nd	6	40	nd	dw	nd	nd	A&R	R	ground	Oceania (Papua New Guinea)	HR-CS-AAS	[32]
0.06 *	Coffee machine	nd	6	40	nd	dw	nd	nd	A	R	ground	Central America (Cuba, Mexico, Honduras, Guatemala)	HR-CS-AAS	[32]
0.05 ± 0.02	Turkish coffee	5	10	200	nd	nd	nd	100	nd	nd	fresh ground	nd	FAAS	[46]
K														
300.85 *	Coffee machine (freeze-dried beverage)	0.83	15	400	300	diw	15	94	A	nd	fine fresh ground (no washed)	nd	INAA	[44]
294.91 *	Coffee machine (freeze-dried beverage)	0.83	15	400	300	diw	15	94	A	nd	fine fresh ground (washed)	nd	INAA	[44]
167.44 *	Coffee machine	nd	6	nd	40	dw	nd	nd	A&R	R	ground	Africa (Kenya)	HR-CS-AAS	[32]
154.07	Aeropress	2	18	nd	250	fw	2–4	93	A	R	coarse ground	nd	ICP-OES	[20]
148.55 *	Coffee machine	nd	6	nd	40	dw	nd	nd	A	R	ground	Asia (China, Timor)	HR-CS-AAS	[32]
140.00	Drip	2.5	18	300	nd	fw	nd	92	A	R	medium coarse ground	nd	ICP-OES	[20]
139 *	Coffee machine (capsules)	nd	5	nd	40	dw	nd	nd	A	R	capsule—ground	South America (Colombia, Brazil)	HR-CS-AAS	[32]
132.69 *	Coffee machine	nd	6	nd	40	dw	nd	nd	A&R	R	ground	Oceania (Papua New Guinea)	HR-CS-AAS	[32]
~120.00	Pouring water	5	17	250	nd	fw	nd	92	A	R	very fine ground	nd	ICP-OES	[20]
119.14 *	Coffee machine	nd	5	nd	40	dw	nd	nd	A&R	R	capsules—ground	Asia (India)	HR-CS-AAS	[32]
110.41 *	Coffee machine	nd	6	nd	40	dw	nd	nd	A	R	ground	Africa (Mussulo and Ethiopia)	HR-CS- AAS	[32]
103.40 *	Coffee machine	nd	6	nd	40	dw	nd	nd	A	R	ground	Central America (Cuba, Mexico, Honduras, Guatemala)	HR-CS-AAS	[32]
~100.00	Coffee machine	nd	17	nd	250	fw	9	92	A	R	very fine ground	nd	ICP-OES	[20]
99.20 ± 17.94	Turkish coffee	5	10	200	nd	nd	nd	100	nd	R	fresh ground	nd	FAAS	[46]
88.74	French pres	5	17	300	nd	fw	1–2	92	A	R	medium ground	nd	ICP-OES	[20]
82.74 *	Pouring water	nd	6	150	nd	nd	nd	hot	A	R	ground	India, Australia, Tanzania, Peru, Cuba, Timor, Zambia, Honduras, Indonesia	FAAS	[42]
82.14 *	Pouring water	nd	6	150	nd	nd	nd	nd	A&R	R	ground	nd	FAAS	[42]
57.03 ± 9.73	Turkish coffee	nd	5	65	nd	up dw	nd	nd	A	R	fine ground	nd	HR-CS-FAAS	[41]
52.20	Pouring water	10	6	150	nd	dw	nd	100	nd	R	ground	nd	FAAS	[43]
37.21	Pouring water	5	6	200	nd	nd	nd	100	nd	R	ground #	Ethiopia	FAAS	[36]
Ca														
43.23 *	Pouring water	10	6	200	nd	rw	nd	100	A	R	ground	nd	FAAS	[31]
37.97 *	Pouring water	10	6	200	nd	rw	nd	100	A&R	R	ground	nd	FAAS	[31]
8.31 *	Coffee machine (freeze-dried beverage)	0.83	15	400	300	diw	15	94	A	nd	fine fresh ground (no-washed)	nd	INAA	[44]
7.81 *	Coffee machine (freeze-dried beverage)	0.83	15	400	300	diw	15	94	A	nd	fine fresh ground (washed)	nd	INAA	[44]
6.78 *	Pouring water	nd	6	150	nd	nd	nd	hot	A	R	ground	India, Australia, Tanzania	FAAS	[42]
5.05 *	Pouring water	nd	6	150	nd	nd	nd	nd	A&R	R	ground	nd	FAAS	[42]
3.49	Pouring water	10	6	150	nd	dw	nd	100	nd	nd	ground coffee	nd	FAAS	[43]
2.57	Coffee machine	nd	17	nd	250	fw	9	92	A	R	very fine ground	nd	ICP-OES	[20]
2.36 *	Coffee machine	nd	6	nd	40	dw	nd	nd	A	R	ground	Africa (Mussulo, Ethiopia)	HR-CS-AAS	[32]
~2.30	Pouring water	5	17	250	nd	fw	nd	92	A	R	very fine ground	nd	ICP-OES	[20]
2.18 *	Coffee machine	nd	5	nd	40	dw	nd	nd	A&R	R	capsules—ground	Asia (India)	HR-CS-AAS	[32]
~2.00	Aeropress	2	18	nd	250	fw	2–4	93	A	R	coarse ground	nd	ICP-OES	[20]
1.94 *	Coffee machine	nd	6	nd	40	dw	nd	nd	A&R	R	ground	Africa (Kenya)	HR-CS-AAS	[32]
~1.70	French press	5	17	300	nd	fw	1–2	92	A	R	medium ground	nd	ICP-OES	[20]
1.63	Drip	2.5	18	300	nd	fw	nd	92	A	R	medium coarse ground	nd	ICP-OES	[20]
1.62	Pouring water	5	6	200	nd	nd	nd	100	nd	nd	ground #	Ethiopia	FAAS	[36]
1.50 *	Coffee machine (capsules)	nd	5	nd	40	dw	nd	nd	A	R	capsule—ground	South America (Colombia, Brazil)	HR-CS-AAS	[32]
1.49 *	Coffee machine	nd	6	nd	40	dw	nd	nd	A&R	R	ground	Oceania (Papua New Guinea)	HR-CS-AAS	[32]
1.45 *	Coffee machine	nd	6	nd	40	dw	nd	nd	A	R	ground	Asia (China, Timor)	HR-CS-AAS	[32]
1.38 ± 0.29	Turkish coffee	5	10	200	nd	nd	nd	100	nd	nd	fresh ground	nd	FAAS	[46]
1.11 *	Coffee machine	nd	6	nd	40	dw	nd	nd	A	R	ground	Central America (Cuba, Mexico, Honduras, Guatemala)	HR-CS-AAS	[32]
Mg														
198.47 *	Pouring water	10	6	200	nd	rw	nd	100	A&R	R	ground	nd	FAAS	[31]
99.60 *	Pouring water	10	6	200	nd	rw	nd	100	A	R	ground	nd	FAAS	[31]
14.92 ± 2.15	Turkish coffee	nd	5	65	nd	dw	nd	nd	A	R	fine ground	nd	HR-CS-FAAS	[41]
12.60 *	Pouring water	nd	6	150	nd	nd	nd	hot	A&R	R	ground	nd	FAAS	[42]
11.63	Aeropress	2	18	nd	250	fw	2–4	93	A	R	coarse ground	nd	ICP-OES	[20]
~10.70	Pouring water	5	17	250	nd	fw	nd	92	A	R	very fine ground	nd	ICP-OES	[20]
~10.00	Drip	2.5	18	300	nd	fw	nd	92	A	R	medium coarse ground	nd	ICP-OES	[20]
8.68 *	Coffee machine	nd	6	40	nd	dw	nd	nd	A&R	R	ground	Africa (Kenya)	HR-CS-AAS	[32]
~8.50	Coffee machine	nd	17	nd	250	fw	9	92	A	R	very fine ground	nd	ICP-OES	[20]
8.39 *	Coffee machine	nd	6	40	nd	dw	nd	nd	A&R	R	ground	Africa (Kenya)	HR-CS-AAS	[32]
8.24 *	Coffee machine	nd	6	40	nd	dw	nd	nd	A&R	R	ground	Oceania (Papua New Guinea)	HR-CS-AAS	[32]
7.72	French press	5	17	300	nd	fw	1–2	92	A	R	medium ground	nd	ICP-OES	[20]
7.49 *	Coffee machine	nd	5	40	nd	dw	nd	nd	A&R	R	capsules-ground	Asia (India)	HR-CS-AAS	[32]
7.03 *	Coffee machine	nd	6	40	nd	dw	nd	nd	A	R	ground	Asia (China, Timor)	HR-CS-AAS	[32]
7.00 ± 1.37	Turkish coffee	5	10	200	nd	nd	nd	100	nd	nd	fresh ground	nd	FAAS	[46]
7.00 *	Coffee machine	nd	6	40	nd	dw	nd	nd	A	R	ground	Africa (Mussulo, Ethiopia)	HR-CS-AAS	[32]
5.35 *	Coffee machine	nd	6	40	nd	dw	nd	nd	A	R	ground	Central America (Cuba, Mexico, Honduras, Guatemala)	HR-CS-AAS	[32]
5.02 *	Pouring water	nd	6	150	nd	nd	nd	hot	A	R	ground	India, Australia, Tanzania, Peru, Cuba, Timor, Zambia, Honduras, Indonesia	FAAS	[42]
2.83	Pouring water	5	6	200	nd	nd	nd	100	nd	nd	ground #	Ethiopia	FAAS	[36]
2.15	Pouring water	10	6	150	nd	dw	nd	100	nd	nd	ground coffee	nd	FAAS	[43]
2.15 *	Turkish coffee	nd	2	100	nd	dw	nd	nd	A	R	nd	nd	ICP-OES	[45]
P														
40.39 *	Coffee machine	nd	6	40	nd	dw	nd	nd	A	R	ground	Africa (Kenya)	HR-CS-AAS	[32]
36.09 *	Coffee machine (capsules)	nd	6	40	nd	dw	nd	nd	A&R	R	ground	Africa (Kenya)	HR-CS-AAS	[32]
33.44 *	Coffee machine	nd	6	40	nd	dw	nd	nd	A&R	R	ground	Oceania (Papua New Guinea)	HR-CS-AAS	[32]
31.95 *	Coffee machine	nd	6	40	nd	dw	nd	nd	A	R	ground	Africa (Mussulo, Ethiopia)	HR-CS-AAS	[32]
27.17 *	Coffee machine	nd	5	40	nd	dw	nd	nd	A&R	R	capsules-ground	Asia (India)	HR-CS-AAS	[32]
27.16 *	Coffee machine	nd	6	40	nd	dw	nd	nd	A	R	ground	Asia (China, Timor)	HR-CS-AAS	[32]
19.44 *	Coffee machine	nd	6	40	nd	dw	nd	nd	A	R	ground	Central America (Cuba, Mexico, Honduras, Guatemala)	HR-CS-AAS	[32]
15.90 *	Pouring water	nd	6	150	nd	nd	nd	hot	A	R	ground	India, Australia, Tanzania, Peru, Cuba, Timor, Zambia, Honduras, Indonesia	FAAS	[42]
13.68 *	Pouring water	nd	6	150	nd	nd	nd	nd	A&R	R	ground	nd	FAAS	[42]
8.16	Aeropress	2	18	nd	250	fw	2–4	93	A	R	coarse ground	nd	ICP-OES	[20]
~7.50	Pouring water	5	17	250	nd	fw	nd	92	A	R	very fine ground	nd	ICP-OES	[20]
~7.20	Drip	2.5	18	300	nd	fw	nd	92	A	R	medium coarse ground	nd	ICP-OES	[20]
~5.60	Coffee machine	nd	17	nd	250	fw	9	92	A	R	very fine ground	nd	ICP-OES	[20]
4.96	French press	5	17	300	nd	fw	1–2	92	A	R	medium ground	nd	ICP-OES	[20]

A—Arabica, A&R—Arabica and Robusta, Av. ± SD—average ± standard deviation, diw—deionized water, dw—distilled water, FAAS—flame atomic absorption spectrometry, fw—filtered water, HR-CS-FAAS—high resolution continuum source flame atomic absorption spectrometry, ICP-OES—inductively coupled plasma optical emission spectrometry, INAA—instrumental neutron activation analysis, nd—no data, R—Robusta, Ro—roasted, up dw—ultrapure distilled water, # mixed five coffee samples of 3 different brands, * the results presented by the authors per weight (100 g or equivalent units).

**Table 2 foods-10-01328-t002:** The content of macroelements per 100 g of instant coffee (no declaration: species, degree of roasting, and origin).

Content Av. (mg/100 g)	Method of Brewing	Time of Brewing (min)	Coffee (g)	Water (mL)	The Volume of a Cup (mL)	Type of Water	Pressure (Ba)	Temperature of Water (°C)	Methods of Analysis	References
Na										
7.14	Pouring water	nd	6	150	nd	nd	nd	hot	FAAS	[42]
1.70	Pouring water	nd	2	nd	30	up dw	nd	hot	HR-CS-FAAS	[23]
K										
283.99	Pouring water	nd	2	nd	30	up dw	nd	hot	HR-CS-FAAS	[23]
50.60	Pouring water	nd	6	150	nd	nd	nd	hot	FAAS	[42]
Ca										
6.96	Pouring water	nd	6	150	nd	nd	nd	hot	FAAS	[42]
1.61	Pouring water	nd	2	nd	30	up dw	nd	hot	HR-CS-FAAS	[23]
Mg										
29.09	Pouring water	nd	2	nd	30	up dw	nd	hot	HR-CS-FAAS	[23]
6.80	Pouring water	nd	6	150	nd	nd	nd	hot	FAAS	[42]
P										
24.78	Pouring water	nd	6	150	nd	nd	nd	hot	FAAS	[42]
7.76	Pouring water	nd	2	nd	30	up dw	nd	hot	HR-CS-FAAS	[23]

Av—average, FAAS—flame atomic absorption spectrometry, HR-CS-FAAS—high resolution continuum source flame atomic absorption spectrometry, nd—no data, up dw—ultrapure distilled water.

## Data Availability

The analyzed publications are available from the authors.

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
