# Peer review of "Coffee Brews: Are They a Source of Macroelements in Human Nutrition?"

_foods, 2021, doi:10.3390/foods10061328_

Round 1

Reviewer 1 Report

The authors provide an impressive review on the mineral content in coffee infusions dependent on various parameters. The main conclusion is that coffee might be a significant source for some minerals.

The following revisions are necessary:

The English must be corrected by a native speaker. There are many instances of Grammar mistakes etc

Title: The title, after the revision, is still somehow confusing and non grammatical.

Title and abstract: it should be mentioned more clearly that only macroelements were mentioned. The abstract should also specify the actual elements under review. Is there still a second part on trace elements planned? This could be more interesting.

Abstract and conclusion: is there a mismatch in minerals. The abstract reports Mg and K as relevant source, while the conclusion reports Na and K as source? I also wonder if there is really anything valuable about Na and K, which are everywhere, they are non-essential and there is clearly not a lack in nutrition. Mg could be more interesting. Would the conclusion not rather be that coffee is irrelevant for mineral intake? Or it just depends on the quality and mineral content of the local water? Sometimes, coffee is also prepared using demineralized water for taste quality reasons. Then there clearly is much lower mineral content?

Line 27: The correct scientific term for robusta would be canephora, not the other way around. Robusta is a variety of canephora. Most commercial coffee sold as robusta is not even Coffea canephora var. robusta, but of other varieties (probably mostly hybrids).

Line 26: please re-check the coffee production data. This looks very low. Wikipedia reports that  Brazil alone is producing more than 2 million metric tons per year. Is there a miscalculation from bags to tons?

General comment: I recently reviewed another paper of the group with the same mistakes. The introduction should be checked for self-plagiarism.

Line 30: spelling diterpenes

Line 40 and throughout: it remains rather vague if coffee itself or the water used for coffee preparation might be the source of minerals? Probably both, and in most studies this confounding cannot be resolved.

Line 88: as these are very rough calculations and estimations with a high variance, would it not have been appropriate to standardize the values with a density of 1?

Reviewer 2 Report

The review “Coffee infusions - are source essential mineral elements in human nutrition?” deals with macroelements present in different coffee brews. In my opinion the paper is interesting and deal with an important aspect of this non-alcoholic beverage. In addition, the current work is the first attempt of review on this field, making it even more eminent.

Specific comments:

- The authors repeated more times “coffee infusion” but generally it is used “coffee brew” or “coffee beverage”. Please replace “coffee infusion” with “coffee brew” or “coffee beverage” in the overall manuscript including the title.

- Line 463: please remove “42”.

- Conclusions: it is too short. Please implement it.
